

# Environmental control of natural gap size distribution in tropical forests

Youven Goulamoussène[1,2], Caroline Bedeau[3], Laurent Descroix[3],
Laurent Linguet[2], and Bruno Hérault[1]

[1]Centre de coopération Internationale de la Recherche Agronomique pour le Développement (CIRAD), UMR EcoFoG, Kourou, French Guiana
[2]Université de Guyane – UMR Espace-Dev, BP 792, 97337 Cayenne, France
[3]Office National des Forêts (ONF),departement Departement RD, Cayenne, French Guiana

*Correspondence to:* Youven Goulamoussène (youven.goulamoussene@ecofog.gf)

**Abstract.** Natural disturbances are the dominant form of forest regeneration and dynamics in unmanaged tropical forests. Monitoring the size distribution of treefall gaps is important to better understand and predict the carbon budget in response to land use and other global changes. In this study, we model the size frequency distribution of natural canopy gaps with a discrete power law distribution. We use a Bayesian framework to introduce and test, using Monte Carlo Markov Chain and Kuo-Mallick algorithms, the effect of local physical environment on gap size distribution. We apply our methodological framework to an original Light Detecting and Ranging dataset in which natural forest gaps were delineated over 30000 ha of unmanaged forest. We highlight strong links between gap size distribution and environment, primarily hydrological conditions and topography, with large gaps being more frequent in floodplains and on hillslopes. In the future, we plan scale up testing of our methodology using satellite data. Additionally, although gap size distribution variation is clearly under environmental control, gap process variation over time should be tested against climate variability.

## 1 Introduction

Natural disturbances caused by forest gaps play an important role in tropical rainforest dynamics. Canopy gaps caused by the death of one or more trees are the dominant form of forest regeneration because the creation of canopy openings continuously reshapes forest structure as gaps are filled with younger trees (Whitmore, 1989). The first, and perhaps most important, effect of gaps is an increase in light intensity (Hubbell et al., 1999a), allowing sunlight to penetrate the understory. This phenomenon has been widely studied because the opening of gaps contributes to the establishment



and growth of light-demanding trees (Denslow et al., 1998), thus contributing to the maintenance of biodiversity. Another effect of canopy gaps is the modification of soil nutrient balance (Rüger et al., 2009). When canopy gaps are created, a large quantity of leaf and wood litter become available, due to high rates of decomposition and mineralization, leading to increased levels of key nutrients (Brokaw, 1985). These nutrients are also linked to spatial variation in forest carbon balance, as shown by Feeley et al. (2007). The relationship between gap formation and the population dynamics of trees or lianas is also quite well understood, with increased liana basal area (Schnitzer et al., 2014) allowing low-wood-density pioneer species to recruit exclusively in newly formed gaps (Molino and Sabatier, 2001).

Many studies have investigated the effect of treefall gaps on biodiversity, particularly animal movement and species composition (Bicknell et al., 2014; Puerta-Piñero et al., 2013), carbon cycles, and forest dynamics. Some authors use field data to study natural gap dynamics, usually at plot scale (Hubbell et al., 1999b). As these studies are quite limited in spatial extent ($< 50$ ha) and because gap formation is largely unpredictable (Hubbell et al., 1999a; Lloyd et al., 2009), optical satellite imagery has been widely promoted and proven adequate for monitoring forest gaps over space and time (Frolking et al., 2009). At high resolution ($< 10$ m), IKONOS satellite images may be well suited for evaluating gap dynamics (Espírito-Santo et al., 2014). In French Guiana, the SPOT$-4$ satellite (20 m spatial resolution) has successfully detected canopy gaps (Colson et al., 2006) using a combination of several spectral bands, such as near and short-wave infrared. However, topographical variation, gap shape, and shade may influence and bias gap detection. Moreover, persistent cloud cover, common in many tropical forest basins, limits the utility of optical satellite data.

Airborne Light Detecting and Ranging (LiDAR) platforms therefore offer a solution to this problem. Recent developments in LiDAR have significantly advanced our ability to derive accurate measurements of canopy forest structure, to detect gaps, and to assess the effect of spatial and temporal variation in carbon balance (Asner and Mascaro, 2014). Kellner and Asner (2009) used remote LiDAR sensing to quantify canopy height and gap size distributions in five tropical rain forest landscapes in Costa Rica and Hawaii. They showed that canopy gaps can be observed with the help of LiDAR-derived digital canopy models (DCMs) and that gap size frequency distribution (GSFD) can be fit with a power law distribution, suggesting a surprising similarity in canopy gap size frequency distributions on diverse soil types associated with diverse geologic substrate ages. Asner et al. (2013) also used LiDAR data to analyze whether gap size frequency distribution is modified by topographic and geologic characteristics and again showed that canopy gap size distribution is largely invariant between forests on erosional terra firme and depositional floodplain substrates in the Peruvian Amazon basin. Finally, using airborne LiDAR, Lobo and Dalling (2014) have recently explored the effect of forest age, topography, and soil type on canopy disturbance patterns across central Panama. For the first time, they highlighted significant effects of slope and of forest age, with a higher frequency of large gaps associated with old-growth forests and gentle slopes.



In this study, we use a DCM derived from airborne LiDAR across a 30000 ha tropical forest landscape in the Régina forest in French Guiana. This approach provides high-resolution maps of
canopy gaps and helps us to understand the environmental determinism of gap occurrence in tropical forests. Our specific aims were therefore:

  – to define canopy gaps from canopy height data using a probabilistic approach

  – to model gap size distribution by inferring a likelihood-explicit discrete power law distribution in a Bayesian framework

– to introduce environment into the scaling parameter of the power law distribution and test its predictive ability

## 2 Materials and Methods

The study site is located in the Régina forest (4°N, 52°W), where the most common soils are ferralitic. The site is located on slightly contrasting plateau-type reliefs that are rarely higher than 150
m on average. The forest is typical of French Guianese rainforests. Dominant plant families in the Régina forest include *Burseraceae*, *Mimosoideae*, and *Caesalpinoideae*. The site receives 3,806 mm of precipitation per year, with a long dry season from mid-August to mid-November, and a short dry season in March (Wagner et al., 2011).

### 2.1 Data source

#### 2.1.1 LiDAR data

LiDAR data were acquired by aircraft in 2013 over 30,000 ha of forest by a private contractor, Altoa (http://www.altoa.fr/), using a Riegl LMS-Q560 laser. This system was composed of a scanning laser altimeter with a rotating mirror; a GPS receiver (coupled to a second GPS receiver on the ground); and an inertial measurement unit to record the pitch, roll, and heading of the aircraft. The laser
wavelength was near-infrared (from about 800 nm to 2500 nm). Flights were conducted at 500 m above ground level with a ground speed of 180 km.h$^{-1}$, and each flight derived two acquisitions. The LiDAR was operated with a scanning angle of 60° and a 200 kHz pulse repetition frequency. The laser recorded the last reflected pulse with a precision better than 0.10 m, with a density of 5 pulses.m$^{-2}$.

The DCM was derived from the raw scatter plot consisting of the pooled dataset from the two acquisitions. Raw data points were first processed to extract ground points using the TerraScan (TerraSolid, Helsinki) ground routine, which classifies ground points by iteratively building a triangulated surface model. Ground points typically made up less than 1% of the total number of the return pulses. The DCM has a resolution of 1 m. A 20 m buffer was applied to the shoreline, so that gaps



closer than 10 m to the shore are not further included in the analyses. A 25 m buffer was applied to anthropogenic tracks to avoid misinterpreting indentations from water and roads in the land as gaps.

### 2.1.2 Environmental data

We use six environmental variables to synthesize the observed environmental gradients. All variables were computed from a LiDAR digital terrain model (DTM) with 5 $m^2$ cells.

### 2.1.3 Slope


The slope was derived from the LiDAR DTM. Slope was computed at a grid cell as the maximum rate of change in elevation from that cell to its 8 neighboring cells over the distance between them.

### 2.1.4 Topographic exposure

We use the TOPEX index to measure topographic exposure to wind (Chapman, 2000). TOPEX is a
variable that represents the degree of shelter assigned to a location. It was derived from quantitative assessment of horizon inclination. The values of this index are closely correlated with wind-shape index (Mikita and Klimánek, 2012). Exposure is calculated based on the height and distance of the surrounding horizon, which are combined to obtain the inflection angle. We use this angle to quantify topographic exposure. When a large topographic feature, like a mountain, is far off in the distance the
inflection angle is low. When the same mountain is closer, the inflection angle is higher. Therefore, a higher inflection angle is equal to lower exposure or higher sheltering (Mikita and Klimánek, 2012).

### 2.1.5 Drained area

Drained area (DA) measures the surface of the hydraulic basin that flows through a cell. A low value indicates that a cell is located at the border between two basins, whereas high values indicate cells
located downstream.

### 2.1.6 Hydraulic altitude

The hydraulic altitude (HA) of each cell, its altitude above the closest stream of its hydraulic basin, was computed from the 3rd order hydraulic system. Low values, including 0, indicate that the forest plot is potentially temporarily flooded, whereas high values indicate that it is located on a hilltop.

### 2.1.7 Terrain ruggedness index

The terrain ruggedness index (TRI) captures the difference between flat and mountainous landscapes. TRI was calculated using SAGA GIS SAGA (2013) as the sum of the altitude change between a pixel and its eight neighboring pixels (Riley, 1999).



### 2.1.8 HAND

The height above the nearest drainage (HAND) model normalizes topography with respect to drainage network by applying two procedures to the DTM. The initial basis for the HAND model came from the definition of a drainage channel: perennial streamflow occurs at the surface, where the soil substrate is permanently saturated. It follows that the terrain at and around a flowing stream must be permanently saturated, independently of the height above sea level at which the channel occurs.

Streamflow indicates the localized occurrence of homogeneously saturated soils across the landscape. The second basis for the HAND model came from the distinctive physical features of water circulation. Land flows proceed from the land to the sea in two phases: in restrained flows at the hillslope surface and subsurface, and in freer flows (or discharge) along defined natural channels. (Nobre et al., 2011)

### 2.2 Forest gap definition

#### 2.2.1 Height threshold

To identify discrete canopy gaps, we had to choose a gap threshold height. Some authors define this threshold at 2 m (Brokaw, 1982). Runkle (1982) defines a gap as the ground area under a canopy opening that extends to the base of the surrounding canopy trees, these usually being considered to

be taller than 10 m, with a trunk diameter at breast height (DBH) > 20 cm. However, in practice, defining gap boundaries is a tricky issue, even in the field. Here, we develop a probabilistic method for detecting canopy gaps from LiDAR data. We used the DCM to model canopy height distribution considering a mixture distribution of two ecological states: the natural variation of canopy height in mature forests, modeled as a normal distribution, and the presence of forest gaps, which lead to a

new normal distribution with lower values. We consider that the threshold between the two states is equal to the $0.001th$ percentile of the height distribution of the canopy. We then define canopy gaps as contiguous pixels at which the vegetation height is less than or equal to the height threshold.

#### 2.2.2 Minimum gap size

In our study, we define the minimum area of a gap as $x_{min}$. We model the gap size frequency

distribution with a power law distribution. We use the Pareto distribution in a discrete power law probability density function (Virkar and Clauset, 2014). These distributions have a negative slope and their size frequencies are plotted on logarithmic axes, allowing us to observe the scaling parameter $\lambda$, which is also called the scaling exponent is not defined when $\lambda \geq 1$. A value close to 1 means there are a large number of small gaps. In a discrete power law with parameter $\lambda$, the probability for

gap size $x$ is given by:



$$p(x) \quad = \quad \frac{x^{-\lambda}}{\zeta(x_{min}, \lambda)}, \tag{1}$$

where $x_{min}$ is the lower truncation point and $\lambda$ is the scaling parameter. The R programming language's poweRlaw package uses the zeta function from the VGAM package to perform this calculation (see Yee et al. (2010)). We use a Kolmogorov-Smirnov (KS) distance criterion order to

determine the error between the observed distribution and the Pareto distribution. KS is defined as the maximum distance between the cumulative distribution functions (CDFs) of the data and the fitted function (Virkar and Clauset, 2014). We retain, for the remainder of this study, a minimum gap size area $x_{min}$ = 104 m$^2$, which minimized the KS distance in our dataset.

### 2.3 Modeling gap size distribution

Having set the height threshold and minimum gap size, the GSFD is modeled with a discrete Pareto distribution frequency.

#### 2.3.1 Model inference

We use a Bayesian framework to estimate model parameters. Here, the value of a parameter is estimated by its posterior distribution, which by definition, is proportional to the product of the

likelihood of the model and the parameter prior distribution. The prior distribution is based on prior knowledge of the possible values of a parameter. The posterior densities of the different parameters were estimated using a Monte Carlo Markov Chain algorithm (MCMC).

#### 2.3.2 Metropolis-Hastings algorithm

As the model contains many parameters, we built a Metropolis-Hastings (MH) algorithm in which

all parameters are updated together. Details on the algorithm are given below:

- $Y = y_1, y_2, \ldots, y_n$ is the gap size vector

- $X = x_{g1}, x_{g2}, \ldots, x_{gi}$ is the vector of covariates (environmental variables) for gap $g$

- $\theta = \theta_1, \theta_2, \ldots, \theta_i$ is the model parameter vector

The first values of the parameter vector are initialized as $t = 1$, $\theta^t \sim \pi_\theta^0$.


For each step $t$, a new parameter value is sampled from the proposition distribution and a new vector of theta candidates is generated.

$$\theta^{cand} \sim \pi^{prop} \tag{2}$$

Acceptance or rejection of the new candidate $\theta^{cand}$ is determined by computing the likelihood

ratio of the two discrete Pareto distributions:





$$\rho(\theta^t, \theta^{cand}) = \underbrace{\frac{\mathcal{L}(Y|X, \theta^{cand})}{\mathcal{L}(Y|X, \theta^t)}}_{\text{likelihood}} \underbrace{\frac{\pi_\theta^0(\theta^{cand})}{\pi_\theta^0(\theta^t)}}_{\text{prior}} \underbrace{\frac{\pi^{prop}(\theta^t)}{\pi^{prop}(\theta^{cand})}}_{\text{proposal}} \tag{3}$$

The candidate $\theta^{cand}$ is accepted or rejected as follows:

$$u \sim \mathcal{U}_{[0,1]}, \theta^{cand} \begin{cases} \theta^{t+1} & if \quad u < \rho(\theta^t, \theta^{cand}) \\ \theta^t & if \quad u > \rho(\theta^t, \theta^{cand}) \end{cases} \tag{4}$$

The algorithm is run for 1000 iterations. We use the median of the posterior densities to estimate parameter values, and the distribution of the posterior densities to estimate parameter credibility intervals.

### 2.3.3 Univariate environmental effects

### 2.3.4 Variable transformation

To improve model inference and parameter significance, we first transform some environmental variables:

$$
\begin{aligned}
Slope &= sqrt(slope) \tag{5} \\
HA &= log(hydraulic\ altitude + 1) \tag{6} \\
TOPEX &= |max(TOPEX) - min(TOPEX)| \tag{7}
\end{aligned}
$$

The environmental variables are then centered and scaled. We first consider each environmental covariate independently. These covariates are included one-by-one in the model to constrain the exponent $\lambda$. We use the exponential function to constrain $\lambda$, because the zeta function is not defined for $\lambda = 1$

$$\lambda_i = 1 + \exp(\theta_{0i} + \theta_i \times var_{ig}) \tag{8}$$

where $\lambda_{ig}$ is the $lambda$ value dependent on the value of environmental variable $i$ in gap $g$, $\theta_{0i}$ is the intercept, and $\theta_i$ quantifies the effect of covariates $var$ on the gap size distribution $var_{ig}$.

### 2.3.5 Multivariate model

### 2.3.6 Principal component analysis

We first investigate the collinearity of environmental data through principal component analysis (PCA) on the normalized environmental dataset.





### 2.3.7 Model

To build the final model, we use the results of the univariate model (Table 1) and the PCA (Figure 3) and set:

$$\lambda = 1 + \exp(\theta_0 + \theta_1 \times Slope + \theta_2 \times TOPEX$$
$$+ \theta_3 \times HA + \theta_4 \times HAND) \tag{9}$$

**Kuo-Mallick**

To select the significant covariates and build the final model, we use the method proposed by Kuo & Mallick (KM) Kuo and Mallick (1998). This method consists of associating an indicator with each variable $var_i$ and parameter $\theta_i$. This indicator can take two values: 1 or 0. If it is set to 1, the variable is included in the model, but if the value is set to 0, it is not. We use the MH and KM algorithms to estimate the indicators $I$ and infer their *a posteriori* distribution in addition to $\theta$.

We start the KM algorithm with $t = 1$, $\theta^t \sim \pi_\theta^0$, $I_j^t \sim \mathcal{B}er(0.5)$ for $j = 1, \ldots, i$). For each covariate $j$ (selected in random order), we use the MH algorithm to update $\theta_j$. To update $I_j$, we compute the ratio $\rho$ (eq10) and generate $I_j^{t+1}$ from a Bernoulli distribution $\mathcal{B}ern(\rho)$:

$$\rho \quad = \quad \frac{1}{1 + \frac{\mathcal{L}(Y|\tilde{X},\theta^t, I_j=0, I_{-j}^t)}{\mathcal{L}(Y|\tilde{X},\theta^t, I_j=1, I_{-j}^t)}} \tag{10}$$

Model inference and data analysis were conducted with R software (R-Core-Team 2012). All maps and geographical information were computed with SAGA (SAGA, 2013) and ArcGIS 10.1.

## 3 Results

### 3.1 Gap delineation

In this study, we used a forest canopy height mixture model to define the maximum height of a given pixel to be included in a forest gap. This probabilistic method produced results that fit the observed canopy height distribution. We retained the 11 m threshold that corresponds to the 0.001th percentile of the canopy height distribution (Figure 1). Given this height, we retained the surface $x_{\min} = 104$ m$^2$ that minimized the KS distance between predictions and observations. Here, our gap definition was therefore defined as an area $> 104$ m$^2$, in which the LiDAR measured canopy height is always $\leq 11$ m.

### 3.2 Basic statistics

We mapped 12,293 gaps with vegetation $\leq 11$ m in height. The mean gap size was 236 m$^2$ with a minimum gap size of 104 m$^2$ and a maximum of 29,063 m$^2$. The total gap area was about 290 ha,



or 1% of the whole surveyed area. The observed gap size distribution was modeled with a Pareto
distribution (Figure 2), leading to a scaling parameter $\lambda_{x_{min}}$ of 2.6.

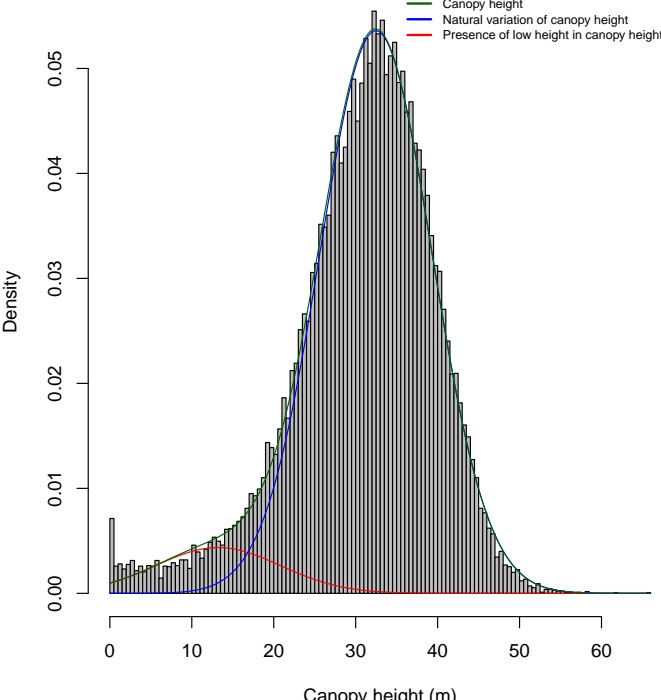

**Figure 1. Canopy height distribution.** Canopy height considered as a mixture distribution of two ecological
features. The first (blue curve) is the natural variation in canopy height, modeled as a normal distribution. The
second (red curve) is linked to the presence of low heights in the total canopy height distribution, likely to be
due to a forest gap. We set the gap threshold to the 0.001th percentile of the blue curve density, *i.e.*, 11 m.

### 3.3 Univariate models

All variables had an effect on gap size distribution (Table 1). The scaling coefficient $\lambda$ is related to
the ratio of small gaps to large gaps, with values close to 1 indicating a higher frequency of large
gaps and vice versa. Parameter estimates for slope and TRI show high occurrence of small gaps for
large values of the two variables. Contrarily, the effect of DA, HAND, HA, and TOPEX on $\lambda$ are
clearly negative, meaning that the frequency of large gaps increases with large values.





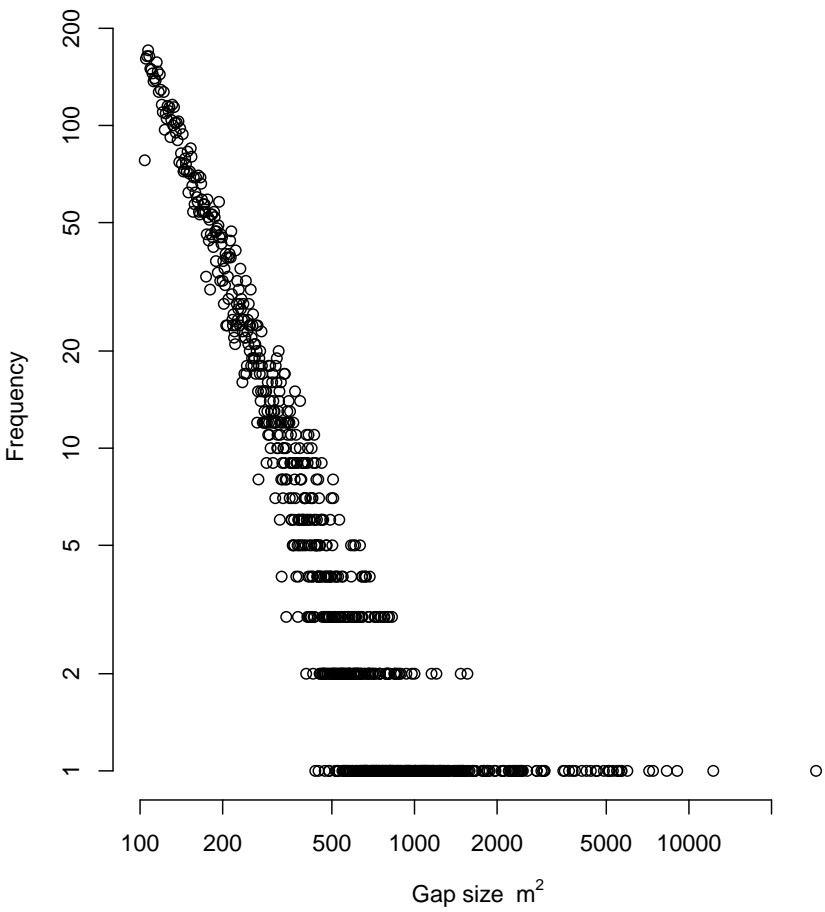

**Figure 2.** The observed gap size frequency distributions modeled as a power law function with $\lambda = 2.6$.

### 3.4 The multivariate model

To define the final multivariate predictive model, we used the significant results of the univariate
models together with the output of the PCA, in order to avoid multicollinearity.

#### 3.4.1 Variable selection

The first three PCA axes explained more than 80% of the data variance. The first axis, which ac-
counted for 36.45% of the variance, was positively correlated with relative HA and negatively corre-
lated to HAND and DA, and thus clearly highlighted the local altitudinal gradient. The second axis



**Table 1.** List of environmental variables, abbreviations, units, and values of the posteriors in univariate models.

| Parameter | Abbreviation | Unit | Posterior value | Confidence interval (CI 95%) |
|---|---|---|---|---|
| Slope | Slope | ° | 0.0735 | [-0.02 ; 0.15] |
| Terrain Ruggedness Index | TRI | - | 0.0718 | [0.04 ; 0.10 ] |
| TOPographic EXposure | TOPEX | - | -0.082 | [-0.12 ; -0.05 ] |
| Drained Area | DA | $m^2$ | -0.0176 | [-0.09 ; 0.05 ] |
| The Hydraulic Altitude | HA | m | -0.0177 | [-0.05 ; 0.02] |
| HAND | HAND | - | -0.003 | [-0.08 ; 0.09 ] |

explained an additional 28.5% of variance and was positively correlated with the TRI and slope. The
third axis explained a further 15.2% of the variance and was correlated only with TOPEX (Figure
3). The multivariate model was created using a Bayesian framework including four environmental
variables: slope, TOPEX, HAND, and HA, the explanatory variables that had an effect on $\lambda$. Finally,
the KM methodological framework was used to select the most parsimonious model.

Environmental covariates with posterior KM values close to 1 Slope, TOPEX, and HAND (eqn
9)were retained in the final model (Figure 4). Parameter estimates of the final model indicated that
the greatest effects on gap size distribution were caused by TOPEX and HAND.

## 4    Discussion

### 4.1    Methodology

#### 4.1.1    Gap Detection

Defining the height threshold at which forest gaps may be delineated is a major difficulty faced by
foresters. Many times, canopy gaps have been defined in the field, adopting Brokaw's definition
(Brokaw, 1982), in which "a 'hole' in the forest extending through all levels down to an average
height of 2 m above the ground," must be defined by an experienced observer. In this study, we have
decided to use a probabilistic approach, modeling height distribution as a mixture of two normal
laws. We found a height, 11 m, which is much higher than that in Brokaw's definition, but is con-
sistent with our field experience, where woody debris, dead canopy tree boles, and residual saplings
(*i.e.*, remnants that survive the gap formation event) may rise well above 2 m. For example, Hubbell
et al. (1999a) showed that small stems frequently remained in gaps up to 4-5 m in height, while
Lieberman et al. (1985) reported broken and damaged stems up to 10 m tall within a gap.

Defining minimum gap size is also a delicate proposition. Some authors, working with high-
definition LiDAR data, have considered a minimum gap size ($x_{min}$) of 1 m$^2$ (Asner et al., 2013)





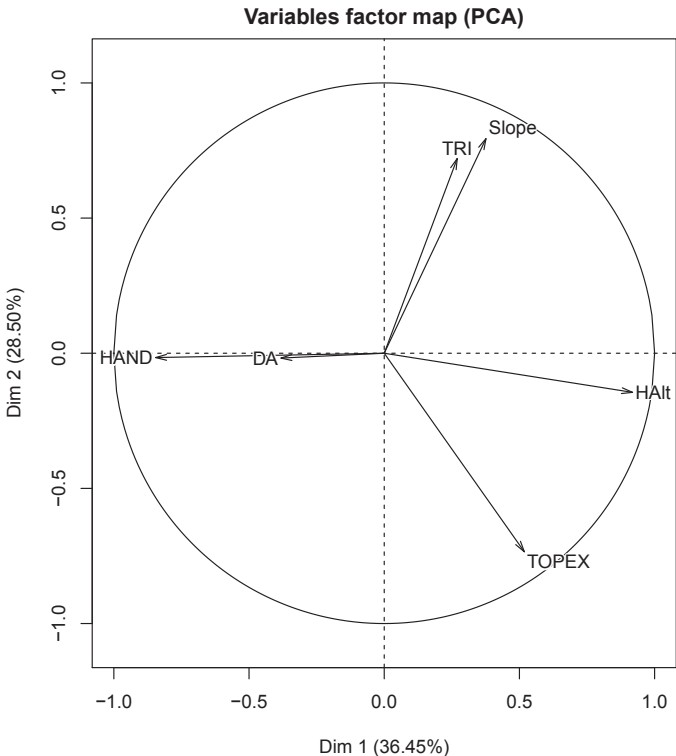

**Figure 3.** Results of the principal component analysis on the environmental variables

(Kellner and Asner, 2009). This minimum gap size is unrealistic from an ecological perspective given that a hole of several square meters in the canopy may simply reflect the distance between two crowns. Brokaw recommended a range from 20 m$^2$ to 40 m$^2$ based on his field experience. We have worked with a minimum gap size of 104 m$^2$, and based this value on the minimized Kolmogorov-





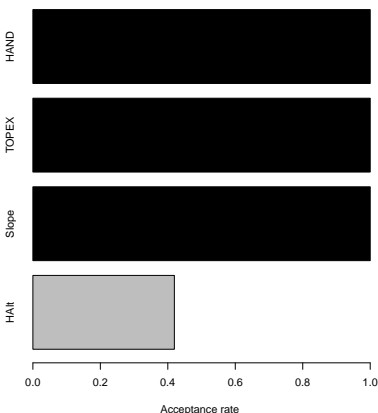

**Figure 4. Results of the Kuo-Mallick algorithm for variable selection.** Variables were included in the final model when their value was close to 100%: $Slope$, $TOPEX$ and $HAND$

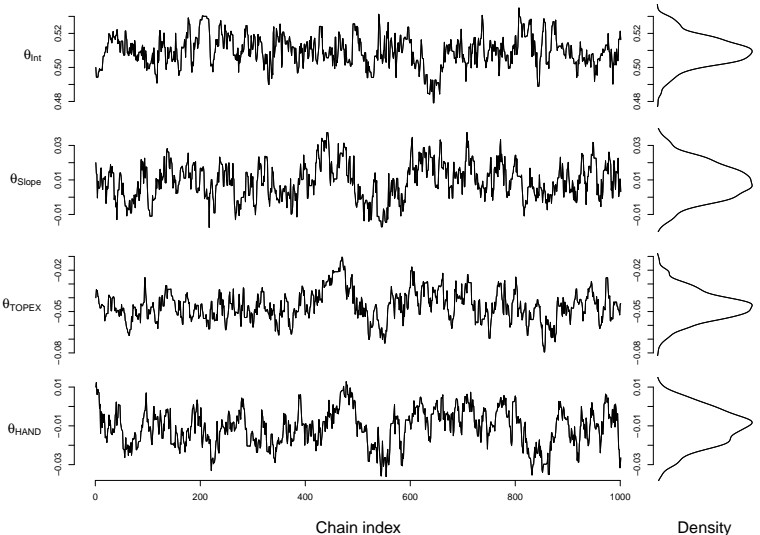

**Figure 5.** Posterior distribution of the environmental variables in the final multivariate model.

Smirnov distance between observed and predicted values. This size is smaller than the average dimensions of a canopy tree crown.



We built on previous studies that show that gap size distribution follows a power law distribution. However, the underlying mechanisms that control this distribution are still unclear. The Bayesian
framework we developed allowed us to detail the contributions of each environmental variable to the size of each individual gap. Because the precise environmental variables were explicitly taken into account in the model likelihood of each gap, we were able to predict gap size distribution from environmental covariates, a difficult task when the scale exponent is estimated once, at the forest level, and compared between forests. The global scale exponent that we estimated for an average
environment ($\lambda = 2.6$), is consistent with some previous studies (Kellner and Asner, 2009; Kellner et al., 2011), though slightly larger than those of others Lobo and Dalling (2014) [1.97 ; 2.15] and Asner et al. (2013) [1.70 ; 2.03].

### 4.2 Environmental effects on gap size frequency distribution

For the first time, gap size distribution integrates environmental variables as a linear combination
of the scale parameter ($\lambda$) of a discrete Pareto distribution frequency. Our results suggest that three covariates drive the gap size frequency distribution in our forest: $Slope$, $HAND$, and $TOPEX$ (Figure 5).

#### 4.2.1 Slope

Steep slopes are well-known to directly impact tropical forest canopy structure (Bianchini et al.,
2010). We found similar results to Lobo and Dalling (2014) in BCI, *i.e.*, large gaps are more frequent on gentle slopes. This may seem counter-intuitive at first, as treefall may be (i) more prone to induce cascading effects when slopes are steep and (ii) more frequent in slopes where soils are shallow with lateral drainage (Gourlet-Fleury et al., 2004), impeding deep rooting of trees. However, uprooting trees account for more than 90% of tree mortality in bottomlands where slopes are gentle
(Durrieu de Madron, 1994). This value drops dramatically to less than 75% on hillslopes and hilltops. Considering that large gaps can be created solely by contiguous treefall, larger gaps may then be expected from a purely probabilistic approach. However, given the positive link between wood density and steep slopes (Ferry et al., 2010), trees may be more resistant to cascading effects than they are in bottomlands.

#### 4.2.2 Water Saturation

$HAND$ is a binary variable that takes the value 1 on water-saturated soils. Because $\lambda$ decreases when $HAND$ equals 1, the frequency of large gaps increases in floodplains and bottomlands. These results support the findings of (Korning and Balslev, 1994), highlighting more dynamic forests in floodplains subject to large flooding events that lead to cascading treefall events. Together with
(Asner et al., 2013), our results suggest that we can effectively extend these results to bottomlands, where we already know that aboveground biomass and mean wood density are 10% lower than on



hilltops (Ferry et al., 2010). Given its ease of implementation on a land-surface model and its high predictive power, HAND covariates present great potential applicability for gap size distribution prediction.

### 4.2.3   Topographic Exposure

The effect of topographic exposure on $\lambda$ is consistent with our *a priori* hypothesis that wind-exposed areas would have a greater relative frequency of large gaps. Although hurricane damage does not occur in continental equatorial regions of the Amazon (Nelson et al., 1994), here we demonstrate that tree exposure has a large impact on gap size distribution. Lobo and Dalling (2014) observed no clear effect of TOPEX, and suggested that this index has a slight negative effect on gap size distribution. The results of this study are in line with the pioneering work of (Negrón-Juárez et al., 2014), which showed that wind exposure is related to higher elevations that inflate the occurrence of larger gaps. However, coastal French Guianese forests exhibit different landscapes and landforms (Guitet et al., 2013). Our study area is made of dissected plateaus characterized by simple forms resembling hills (Guitet et al., 2013). It is possible that these characteristics, leading to unique combinations of landform elevations, may create complex terrain interactions that increase wind local speed and, in turn, cause large gaps. We conclude that topographic exposure is an appropriate index for predicting gap size distribution, but this must be confirmed in other landscape types.

## 5   Conclusions

To our knowledge, this is the first study to demonstrate the effect of environment on forest gap size distribution. We first put forward an innovative method to define a height threshold and minimum gap size using two probabilistic approaches. The modeled distribution of canopy height as mixture of two distributions provides a clear height threshold, while the minimization of KS distance between observed and predicted data proves to be efficient for setting the minimum gap size. We use a Bayesian framework in which the model likelihood of each gap is expressed as a function of the unique environment local to the gap, highlighting the predominant role of the topographic exposure and waterlogging in determining gap size distribution. We expected that slope would also play an important role, with steeper slopes leading to larger gap sizes, but found the opposite effect, as already highlighted by (Lobo and Dalling, 2014). We suggest that our modeling approach can be a basis for the development of large-scale methodologies using satellite data to understand gap phase dynamics at a regional scale, combining LiDAR and RaDAR remote sensing tools.

*Acknowledgements.* B.H. was supported by a grant from the Investing for the Future program (managed by the French National Research Agency (ANR, labex CEBA, ref. ANR-10-LABX-0025).





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
