# Peer review of "Environmental control of natural gap size distribution in tropical forests"

_Biogeosciences, 2016_

## Referee Comment (RC1) · Ervan Rutishauser (Referee) · 12 Oct 2016

**Ervan Rutishauser (Referee)**

er.rutishauser@gmail.com

Received and published: 12 October 2016

General comments : Y. Goulamoussène and collaborators are presenting an original study aiming at characterizing canopy gaps in tropical forests using a novel analytic approach. Generally, this study provides new and interesting insights in key environmental drivers of gap formation at landscape scale. While this study certainly deserves to be published, I have pointed a few issues that should be addressed before publication. The authors have developed an innovative analytic method to define gaps, but the entire analysis relies upon an a priori threshold equal to the 0.001th percentile of the estimated "natural variation of canopy height". While this choice may be well grounded, the rationale beyond it remains unexposed. How sensitive is the definition of gap and all subsequent results to this threshold? What if the authors had chosen the 0.01th percentile? Some kind of sensitivity analysis would make their choice more reliable.

More importantly, while a landscape scale approach seems meaningful to infer gap size distribution, this study highlights the importance of environmental factors on both gap frequency and size. Thus, I wonder if a fixed definition of gap remains meaningful, or if that definition should not adapt to the different forest types and/or main topographical features found at large scale. Doing so would point towards a more "ecological" definition of gaps, instead of a pure remote-sensing approach, and ultimately raises the question of the aim of detecting gaps. For instance, does a 100m2 gap in waterlogged areas dominated by Euterpe oleracea has the same ecological meaning than on hill-tops? Certainly not in term of number of trees killed, biomass loss and forest turnover . Depending on the variable of interest (e.g. carbon emission), a fit-them-all definition is questionable. This issue, if not formally addressed, should at least be discussed. The following recent publications may provide additional information (Chambers et al., 2013; Lobo and Dalling, 2014; Schliemann and Bockheim, 2011).

Finally, the manuscript requires additional efforts in editing (loads of typo &citations errors, unclear headers and acronyms) and reviewing recent literature (lots of relevant publications is missing, comparing lambda with other studies). A proof-reading by a native English would also greatly help.

Hope this help. Regards, Ervan Rutishauser

Minor comments : I. 99: For clarity, please define explicitly all the acronyms used, i.e. topographic exposure (TOPEX). I.119: Sub-header should be : "Height above the nearest drainage" to be consistent with previous sub-headers I.193: What is the resolution of the TOPEX variable? Do you have several indices by 5m2? Please clarify how you can get 2 values (min max), or did you standardize TOPEX as: abs((TOPEX – min(TOPEX))/(max(TOPEX)-min(TOPEX))). I. 212: I suggest to change the header here, as Kuo-Mallik refers to a method, but you used it to select the variables. "Variables selection" looks more appropriate. There is also an issue in the way the reference is quoted. I. 216: there is an missing (or extra) parenthesis in your expression I. 226: "Given this height, we retained the surface xmin = 104 m2". What is the link between

BGD
the height threshold and the minimal gap area, here? I thought both minimal height and gap size were defined separately. I.254: "Environmental covariates with posterior KM values close to 1, NAMELY Slope, TOPEX, and HAND ... " I.260: "Defining the height threshold at which forest gaps may be delineated is a major difficulty faced by foresters. Many times, canopy gaps have been defined in the field, adopting Brokaw's definition" Is it only the minimal height that is at stake here, or also the minimal area? Many studies define gaps regarding to their size (e.g. Denslow et al., 1998; Hérault et al., 2010; Lima, 2004). This sentence sounds odd, I suggest rephrasing as follow: "Delineating forest gaps is a persistent challenge for foresters and ecologists, among whom Brokaw's gap definition (1982) has remained very popular/extensively used." I. 265-269: There are several studies that do not use this 2m-threshold definition of gaps, but 10m (e.g. Hubbell et al., 1999; Meer and Bongers, 1996; Welden et al., 1991). While the authors are extensively referring to the seminal paper of Brokaw, there are way more references defining gaps in complex tropical forests that are lacking here. 1.278: Which ones? 1.300 (onwards): This paragraph is very confusing. Where does the 75% comes from? What is the remain 25% then? In sloppy areas, does it make a big difference if a tree falls due to breakage, or being uprooted? I don't think so, and tree size seems to be a more important factor in the cascading effect than mode of death. Yet, the turnover may be more rapid on slopes than bottomland, resulting in fewer large trees (and tree fall gaps). I.338: but WE found

Potentially useful references :

Chambers, J. Q., Negron-Juarez, R. I., Marra, D. M., Di Vittorio, A., Tews, J., Roberts, D., Ribeiro, G. H. P. M., Trumbore, S. E. and Higuchi, N.: The steady-state mosaic of disturbance and succession across an old-growth Central Amazon forest landscape, Proc. Natl. Acad. Sci., 110, 3949–3954, 2013. Colson, F., Gond, V., Freycon, V., Bogaert, J. and Ceulemans, R.: Detecting natural canopy gaps in Amazonian rainforest, Bois For. Trop., 288, 69–80, 2006. Denslow, J. S., Ellison, A. M. and Sanford, R. E.: Treefall gap size effects on above- and below-ground processes in a tropical wet
forest, J. Ecol., 86, 597–609, 1998. Ferry, B., Morneau, F., Bontemps, J. D., Blanc, L. and Freycon, V.: Higher treefall rates on slopes and waterlogged soils result in lower stand biomass and productivity in a tropical rain forest, J. Ecol., 98, 106-116, 2010. Hérault, B., Ouallet, J., Blanc, L., Wagner, F. and Baraloto, C.: Growth responses of neotropical trees to logging gaps, J. Appl. Ecol., 47, 821-831, 2010. Hubbell, S. P., Foster, R. B., O'Brien, S. T., Harms, K. E., Condit, R., Wechsler, B., Wright, S. J. and De Lao, S. L.: Light-gap disturbances, recruitment limitation, and tree diversity in a neotropical forest, Science, 283, 554, 1999. Lima, R. A. F.: Gap size measurement: the proposal of a new field method, For. Ecol. Manag., 214, 413-419, 2004. Lobo, E. and Dalling, J. W.: Spatial scale and sampling resolution affect measures of gap disturbance in a lowland tropical forest: implications for understanding forest regeneration and carbon storage, Proc. R. Soc. Lond. B Biol. Sci., 281(1778), doi:10.1098/rspb.2013.3218, 2014. Meer, P. J. van der and Bongers, F.: Formation and closure of canopy gaps in the rain forest at Nouragues, French Guiana, Vegetatio, 126(2), 167–179, doi:10.1007/BF00045602, 1996. Schliemann, S. A. and Bockheim, J. G.: Methods for studying treefall gaps: A review, For. Ecol. Manag., 261, 1143-1151, doi:10.1016/j.foreco.2011.01.011, 2011. Welden, C. W., Hewett, S. W., Hubbell, S. P. and Foster, R. B.: Sapling Survival, Growth, and Recruitment: Relationship to Canopy Height in a Neotropical Forest, Ecology, 72(1), 35-50, doi:10.2307/1938900, 1991.

---

## Referee Comment (RC2) · M. Bauters (Referee) · 17 Oct 2016

In this manuscript Goulamoussène et al. report on an interesting analysis on the environmental drivers behind gap size distribution in tropical forests, using Lidar data. Their methodology builds on previous work by Lobo and Dalling (2014), but they introduce an interesting new method to determine height thresholds and minimum gap sizes. They rightfully criticize the empirical cutoffs that have been used, and come up with an innovative and interesting alternative. The authors further introduce Lidar-derived environmental parameters to include in the Bayesian model, which enables them to assess the effect of physical environment on gap size distribution.

Although I've enjoyed reading this interesting work, I think the authors could improve/clarify their manuscript in some parts; both for science and form/language.

[Figure]

A first point would be the height cutoff; the whole study is dependent on the value you set here. You set up a nice probabilistic model, but then you still go back to an empirical threshold (being 0,001percentile). If you have no theoretical considerations to justify this cutoff, than your method is as such not better than any other empirical threshold used in previous studies. Hence, this would need bit more clarification. I wonder whether you could not use the average of the second (lower) normal distribution as a cutoff. This would, I believe, also increase the applicability of your method in other forest types/regions, since the relative difference between canopy tree height/gap height will shift in other forest types. Even if you have a sound reason to use 0,001perc in this case, you would need to reselect a threshold in other forest types.

Secondly, I have not been working with Bayesian statistics myself, but I think the manuscript should be clear for the broad readership of Biogeosciences. Some questions related to the rest of your methodology: -I wonder why you use equation 8 to constrain lambda. What is the reasoning behind an exponential model? -You explain the interpretation of Lambda on p 5 L 146-150. I think the authors are confused here (or maybe I am...); "lambda is not defined when lambda>=1?? ", and "A value close to 1 means there are a large number of small gaps". Both of these statements seem absolutely wrong to me (I would expect the contrary with both), and they are actually vital to the interpretation of this very manuscript. Either I am wrong, but if not, I am a bit worried for the misinterpretation of the results by the authors. Please have a look at this. Maybe this is also at the basis for the contradiction in some of the statements through the manuscript; Abstract (L10); "... with large gaps being more frequent on hillslopes". Discussion (L296-297): ""We found similar results to Lobo and Dalling in BCI; i.e., large gaps are more frequent on gentle slopes". Conclusion: "We expected that slope would also play an important role, with steeper slopes leading to larger gap sizes, but found the opposite effect." Please go through the MS again and make sure the interpretations are the same, and are right, everywhere. If not, you fail to give the reader a clear take home message...

Thirdly, I have listed some other comments. The list is not complete; some of these are clear typos or sloppiness. This would need to be avoided for your resubmission. . . In general; make sure results, M and M, and discussion are in the appropriate section, redo your subheadings, avoid typos, avoid repetition. . .

-P1 L10: "we plan to scale up"

-P2 L23: a large quantity of leaf and wood litter becomes available. But please rephrase this anyway. It's not a good sentence. Mineralisation and decomposition makes the nutrients available, not the wood and leaf litter available as such.

-p4 L90: the buffer you applied to anthropogenic tracks: is the Approuage an anthropogenic track? For sure not masking out natural rivers out of your algorithm would hugely affect your results. I think (hope) you did include natural rivers in your buffers, but you would need to rephrase, since these are not anthropogenic. . .

P4: Sloppiness; your subtitles have the same rank on this page, while they all fall within the first subtitle 'Environmental data'

P5 L 141-142: what do you define as contiguous? Diagonal pixels would be contiguous? You know from the field that some trees may be left standing in certain gaps, so this could be important for your results. . .

P6 l 153-154: Please cite both R packages properly.

P6 L172: Here you use X as the vector of covariates; while on the next page in equation 8 you use varig. Please be consistent to make your MS more comprehensible.

P7 L191-193: For clarity I would rename the transformed variables. Also in eq 6 you use the HA as the new variable, and the hydraulic altitude in full as the old, while in 2.1.6. on p 4 you already use the HA abbreviation. Please correct these small errors for your future readers.

P7 l 204: investigated; Material and methods should be in past tense. Please correct

everywhere.

Figure 4: This figure does not have a lot of information. I would leave it out and describe in text instead.

P14 L285: why don't you show the values from the Kellner studies in brackets, like you do for Lobo and Asner?

P14 L300: Please add reference to your 75% statement. . .

P 14 l 309-311: "Together with. . ." I don't get this sentence. Please rephrase. . .

P 15 l 330-331: Really? The first study? And what about Lob and Dallin (a study which served as a basis for your study)

Please review the reference list.

---

## Author Comment (AC1) · 23 Nov 2016

Dear Editor,

Please find herewith our revised ms "Environmental control of natural gap size distribution in tropical forests" for re-submission to Biogeosciences. We first would like to thank both reviewers for the high quality and seriousness of their review. We acknowledge them for that and we recognize that their works have contributed to improve the quality of our manuscript. Responses to their comments are in red in the following as well as are changes in the main ms.

We remain at your disposal for any further enquiries.

Please also note the supplement to this comment (Responses to the reviewers)

Youven Goulamoussène,

Please also note the supplement to this comment:
http://www.biogeosciences-discuss.net/bg-2016-320/bg-2016-320-AC1-
supplement.pdf

―――――――――――――――――――――

**Supplement:**

**Reviewer 1**

Generally, this study provides new and interesting insights in key environmental drivers of gap formation at landscape scale. While this study certainly deserves to be published, I have pointed a few issues that should be addressed before publication. The authors have developed an innovative analytic method to define gaps, but the entire analysis relies upon an a priori threshold equal to the 0.001th percentile of the estimated "natural variation of canopy height". While this choice may be well grounded, the rationale beyond it remains unexposed. How sensitive is the definition of gap and all subsequent results to this threshold ? What if the authors had chosen the 0.01th percentile ? Some kind of sensitivity analysis would make their choice more reliable.

We easily admit that our initial choice of the 0.001th percentile for the height threshold may have been seen as arbitrary. We choose this threshold value to keep the maximum of information from the first distribution (gap height) while minmizing biases due to including too much information from the 2nd one (canopy height). Because the whole process needs several hours to be performed, from the GIS works to the model inference, with thousands of gaps, we cannot run a well-performed sensitivity analysis. However, we provide in supplementary informations , the parameter values for 2 additional thresholds (0.0001th and 0.01th). The posterior values of almost all variables are quite similar and thus do not change interpretation : Slope, TRI are always positive whatever the threshold TOPEX, HAND are always negative whatever the threshold DA and HAlt always include zero in the credibility interval

More importantly, while a landscape scale approach seems meaningful to infer gap size distribution, this study highlights the importance of environmental factors on both gap frequency and size. Thus, I wonder if a fixed definition of gap remains meaningful, or if that definition should not adapt to the different forest types and/or main topographical features found at large scale. Doing so would point towards a more "ecological" definition of gaps, instead of a pure remote-sensing approach, and ultimately raises the question of the aim of detecting gaps. For instance, does a 100m2 gap in waterlogged areas dominated by Euterpe oleracea has the same ecological meaning than on hilltops ? Certainly not in term of number of trees killed, biomass loss and forest turnover . Depending on the variable of interest (e.g. carbon emission), a fit-them-all definition is questionable. This issue, if not formally addressed, should at least be discussed. The following recent publications may provide additional information (Chambers et al., 2013 ; Lobo and Dalling, 2014 ; Schliemann and Bockheim, 2011).

The choice of the values of height and threshold may be adapted to different forest types and topographic characteristics. In our case, the choice was fully data-driven using the DCM and DEM and no ecological knowledge. Within our framework it is likely that in waterlogged areas, areas covered with mature trees that do not exceed the height thresholds may appear in our analysis as forest gaps. In order to clarify this question, an approach using time-series would allow to identify these 'false' gaps that never get filled and thus are not part of the forest endogeneous dynamics. These are not gaps in the ecological meaning.

Finally, the manuscript requires additional efforts in editing (loads of typo citations errors, unclear headers and acronyms) and reviewing recent literature (lots of relevant publications is missing, comparing lambda with other studies). A proof-reading by a native English would also greatly help. This has been done. The manuscript has been edited by a professional science editing service. We do believe that directly comparing lambda values between studies is difficult because it may depend on the assumed (or inferred) height and size thresholds.

**Specific comments**

— l. 99 : For clarity, please define explicitly all the acronyms used, i.e. topographic exposure (TOPEX).
See line 99

— l.119 : Sub-header should be : "Height above the nearest drainage" to be consistent with previous sub-headers
See line 120 . Section Methods 2.1.2.

— l.193 : What is the resolution of the TOPEX variable ? Do you have several indices by 5m2 ? Please clarify how you can get 2 values (min max), or did you standardize TOPEX as : abs((TOPEX – min(TOPEX))/(max(TOPEX)-min(TOPEX))).
The native pixel resolution is 5 m× 5m. Original values for TOPEX are hardly interpretable because they are counter-intuitive. In order to simplify the interpretation, we thus modified the TOPEX values to get the highest for the highest exposure. See the variable trasnformation section.

— l. 212 : I suggest to change the header here, as Kuo-Mallik refers to a method, but you used it to select the variables. "Variables selection" looks more appropriate. There is also an issue in the way the reference is quoted.
Line 212 : Agreed

— l. 216 : there is an missing (or extra) parenthesis in your expression
Line 217 : done

— l. 226 : "Given this height, we retained the surface xmin = 104 m2". What is the link between the height threshold and the minimal gap area, here ? I thought both minimal height and gap size were defined separately.
Yes they are. Firstly, we define a height threshold from which we observe a gap. Secondly we used this height threshold in order to determine the minimum gap size area using the Kolmogorov-Smirnov (KS) distance criterion

— l.254 : "Environmental covariates with posterior KM values close to 1 , NAMELY Slope, TO-PEX, and HAND ...
Line 254 : Done. "Environmental covariates with posterior KM values close to 1, namely Slope, Topex, and HAND"

— l.260 : "Defining the height threshold at which forest gaps may be delineated is a major difficulty faced by foresters. Many times, canopy gaps have been defined in the field, adopting Brokaw's definition" Is it only the minimal height that is at stake here, or also the minimal area ? Many studies define gaps regarding to their size (e.g. Denslow et al., 1998 ; Hérault et al., 2010 ; Lima, 2004). This sentence sounds odd, I suggest rephrasing as follow : "Delineating forest gaps is a persistent challenge for foresters and ecologists, among whom Brokaw's gap definition (1982) has remained very popular/extensively used."
We thank you for your suggestion. The changes were made at line 263

— l. 265-269 : There are several studies that do not use this 2m-threshold definition of gaps, but 10m (e.g. Hubbell et al., 1999 ; Meer and Bongers, 1996 ; Welden et al., 1991). While the authors are extensively referring to the seminal paper of Brokaw, there are way more references defining gaps in complex tropical forests that are lacking here.
We have updated the bibliography

— l.300 (onwards) : This paragraph is very confusing. Where does the 75% comes from ? What is the remain 25% then ? In sloppy areas, does it make a big difference if a tree falls due to breakage, or being uprooted ? I don't think so, and tree size seems to be a more important factor in the cascading effect than mode of death. Yet, the turnover may be more rapid on slopes than bottomland, resulting in fewer large trees (and tree fall gaps).
Rewritten

— l.338 : but WE found
Done, line 345

---

## Author Comment (AC2) · 23 Nov 2016

Dear Editor,

Please find herewith our revised ms "Environmental control of natural gap size distribution in tropical forests" for re-submission to Biogeosciences. We first would like to thank both reviewers for the high quality and seriousness of their review. We acknowledge them for that and we recognize that their works have contributed to improve the quality of our manuscript. Responses to their comments are in red in the following as well as are changes in the main ms.

We remain at your disposal for any further enquiries.

Youven Goulamoussène,

[Figure]

Please also note the supplement to this comment:
http://www.biogeosciences-discuss.net/bg-2016-320/bg-2016-320-AC2-
supplement.pdf

──────────────────────────

**Supplement:**

**1 Reviewer 2**

In this manuscript y et al. report on an interesting analysis on the environmental drivers behind gap size distribution in tropical forests, using Lidar data. Their methodology builds on previous work by Lobo and Dalling (2014), but they introduce an interesting new method to determine height thresholds and minimum gap sizes.

They rightfully criticize the empirical cutoffs that have been used, and come up with an innovative and interesting alternative. The authors further introduce Lidar-derived environmental parameters to include in the Bayesian model, which enables them to assess the effect of physical environment on gap size distribution.

Although I've enjoyed reading this interesting work, I think the authors could im prove/clarify their manuscript in some parts; both for science and form/language

A first point would be the height cutoff; the whole study is dependent on the value you set here. You set up a nice probabilistic model, but then you still go back to an empirical threshold (being 0,001percentile). If you have no theoretical considerations to justify this cutoff, than your method is as such not better than any other empirical threshold used in previous studies. Hence, this would need bit more clarification. I wonder whether you could not use the average of the second (lower) normal distribution as a cutoff. This would, I believe, also increase the applicability of your method in other forest types/regions, since the relative difference between canopy tree height/gap height will shift in other forest types. Even if you have a sound reason to use 0,001perc in this case, you would need to reselect a threshold in other forest types.

Thank you for this comment. If the threshold is chosen from the first distribution, then the expected value of the first law will be too dependant on the frequency of gap formation in time and this dependence is a problem. In other words, with a similar gap generation process (the same drop in height) a frequency of 1% of the forest area that will be transformed in gaps each year will generate a target value (the average of the first law as suggested) mechanically lower than for a landscape where the frequency is 0.5%. In the last case, the lower frequency of young gaps will produce on average a higher target value. We still do not want to depend on this gap frequency at the landscape scale.
We choose this threshold value to keep the maximum of information from the first distribution (gap height) while minmizing biases due to including too much information from the 2nd one (canopy height). Because the whole process needs several hours to be performed, from the GIS works to the model inference, with thousands of gaps, we cannot run a well-performed sensitivity analysis. However, we provide in supplementary informations , the parameter values for 2 additional thresholds (0.0001th and 0.01th). The posterior values of almost all variables are quite similar and thus do not change interpretation : Slope, TRI are always positive whatever the threshold TOPEX, HAND are always negative whatever the threshold DA and HAlt always include zero in the credibility interval

Secondly, I have not been working with Bayesian statistics myself, but I think the manuscript should be clear for the broad readership of Biogeosciences. Some questions related to the rest of your methodology :

— I wonder why you use equation 8 to constrain lambda. What is the reasoning behind an exponential model?

We have constrained the value of lambda in equation 8 because the linear combination, without the exponential constraint, may have result, during the inference process, in negative lambda values. And Riemann's Zeta function only admits values $> 1$.

— You explain the interpretation of Lambda on p 5 L 146-150. I think the authors are confused here (or maybe I am...); "lambda is not defined when lambda$>=$1 ?? ", and "A value close to 1 means there are a large number of small gaps". Both of these statements seem absolutely wrong to me (I would expect the contrary with both), and they are actually vital to the interpretation of this very manuscript. Either I am wrong, but if not, I am a bit worried for the misinterpretation of the results by the authors. Please have a look at this. Maybe this is also at the basis for the contradiction in some of the statements through the manuscript;

I apologize for this mistake in the manuscrit. Thank you for pointing it out. The error has been corrected at line : "In forests dominated by small canopy openings, values of $\lambda$ are larger, whereas smaller values of $\lambda$ increase the frequency of large events (Fisher et al. 2008)". Line : 147

— Discussion (L296-297) : ""We found similar results to Lobo and Dalling in BCI; i.e., large gaps are more frequent on gentle slopes". Conclusion : "We expected that slope would also play an important role, with steeper slopes leading to larger gap sizes, but found the opposite effect." Please go through the MS again and make sure the interpretations are the same, and are right, everywhere. If not, you fail to give the reader a clear take home message

the text has been modified for each section. Line 10, 305, 345

Thirdly, I have listed some other comments. The list is not complete; some of these are clear typos or sloppiness. This would need to be avoided for your resubmission ... In general; make sure results, M and M, and discussion are in the appropriate section, redo your subheadings, avoid typos, avoid repetition...

**Specific comments**

— P1 L10 : "we plan to scale up"
  done line 10
— P2 L23 : a large quantity of leaf and wood litter becomes available. But please rephrase this anyway. It's not a good sentence. Mineralisation and decomposition makes the nutrients available, not the wood and leaf litter available as such.
  Rewritten. Line 24
— p4 L90 : the buffer you applied to anthropogenic tracks : is the Approuage an anthropogenic track? For sure not masking out natural rivers out of your algorithm would hugely affect your results. I think (hope) you did include natural rivers in your buffers, but you would need to rephrase, since these are not anthropogenic...
  Line 88 : Approuague is a large natural river. Indeed, we have applied a buffer in order to only account for areas (i) not affected by forest logging and (ii) natural rivers : *In order to remove areas close to natural rivers : a 20 m buffer was first applied to all shorelines. Then a 25 m buffer was applied to anthropogenic tracks.*
— P4 : Sloppiness; your subtitles have the same rank on this page, while they all fall within the first subtitle 'Environmental data' Page 4 : Paragraphs are now correctly ranked. See the document Line 96
— P5 L 141-142 : what do you define as contiguous? Diagonal pixels would be contiguous? You know from the field that some trees may be left standing in certain gaps, so this could be important for your results . Line 144 : contiguous is defined as a pixel that has any contact with another i.e., a contact by edges or by vertices. In our framework, diagonal pixels are indeed considered as contiguous pixels.
— P6 l 153-154 : Please cite both R packages properly. Done. Line 157 : Most of the analysis was performed under R and making use of poweRlaw and VGAM packages.
— P6 L172 : Here you use X as the vector of covariates; while on the next page in equation 8 you use varig. Please be consistent to make your MS more comprehensible. The vector of covariates as been properly replaced as X in equation 8.
— P7 L191-193 : For clarity I would rename the transformed variables. Also in eq 6 you use the HA as the new variable, and the hydraulic altitude in full as the old, while in 2.1.6. on p 4 you already use the HA abbreviation. Please correct these small errors for your future readers.
  Done. Transformed variables have a new name. Line 195 : Halt, Topex
— P7 l 204 : investigated; Material and methods should be in past tense. Please correct
  Done. Line : 207
— Figure 4 : This figure does not have a lot of information. I would leave it out and describe in

text instead. We prefer maintaining this fig in the text but remains open to suggests from the editor.
— P14 L285 : why don't you show the values from the Kellner studies in brackets, like you do for Lobo and Asner ?Line 295, Done
— P14 L300 : Please add reference to your 75% statement. Line 306 : We changed this sentence
— P 14 l 309-311 : "Together with : : :" I don't get this sentence. Please rephrase : : : Line 320 : In agreement with Asner et al., (2013), our results suggest that we can effectively extend these results to bottomlands, where we already know that aboveground biomass and mean wood density are 10% lower than on hilltops (Ferry et al., 2010).
— P 15 l 330-331 : Really ? The first study ? And what about Lob and Dallin (a study which served as a basis for your study) Please review the reference list. Line 340 : Rewritten. "To our knowledge, this is the first study where the precise environmental descriptors associated to each canopy gap were explicitly taken into account in the general model likelihood. We were able to do so because we wrote general model likelihood as the product of all the single likelihoods (*i.e.* each gap had its own likelihood depending on the environmental covariate values). Doing so, we were able to predict gap size distribution from the fine environmental covariates, an impractical task when the scale exponent is estimated once at the forest level (*i.e.* mixing all the found gaps together) and compared between forests *a posteriori*"

**2 Supplementary information**

TABLE 1 – List of environmental variables, abbreviations, units, and values of the posteriors in univariate models for a height threshold equal to the $0.0001th$ percentile of the height distribution of the canopy.

| Parameter | Abbreviation | Unit | Posterior value | Confidence interval (CI 95%) |
|---|---|---|---|---|
| Slope | Slope | ° | 0.119 | [0.0416 ; 0.208] |
| Terrain Ruggedness Index | TRI | - | 0.119 | [0.083 ; 0.157 ] |
| TOPographic EXposure | TOPEX | - | -0.128 | [-0.188 ; 0.00202 ] |
| Drained Area | DA | $m^2$ | 0.0843 | [-0.0574 ; 0.179 ] |
| The Hydraulic Altitude | HAlt | m | -0.0135 | [-0.04 ; 0.042] |
| HAND | HAND | - | -0.0615 | [-0.152 ; 0.0162] |

TABLE 2 – List of environmental variables, abbreviations, units, and values of the posteriors in univariate models for a height threshold equal to the $0.001th$ percentile of the height distribution of the canopy.

| Parameter | Abbreviation | Unit | Posterior value | Confidence interval (CI 95%) |
|---|---|---|---|---|
| Slope | Slope | ° | 0.0735 | [-0.02 ; 0.15] |
| Terrain Ruggedness Index | TRI | - | 0.0718 | [0.04 ; 0.10 ] |
| TOPographic EXposure | TOPEX | - | -0.082 | [-0.12 ; -0.05 ] |
| Drained Area | DA | $m^2$ | -0.0176 | [-0.09 ; 0.05 ] |
| The Hydraulic Altitude | HAlt | m | -0.0177 | [-0.05 ; 0.02] |
| HAND | HAND | - | -0.003 | [-0.08 ; 0.09 ] |

TABLE 3 – List of environmental variables, abbreviations, units, and values of the posteriors in univariate models for a height threshold equal to the $0.01th$ percentile of the height distribution of the canopy.

| Parameter | Abbreviation | Unit | Posterior value | Confidence interval (CI 95%) |
|---|---|---|---|---|
| Slope | Slope | ° | 0.0975 | [-0.02 ; 0.17] |
| Terrain Ruggedness Index | TRI | - | 0.089 | [0.05 ; 0.12 ] |
| TOPographic EXposure | TOPEX | - | -0.012 | [-0.03 ; -0.32 ] |
| Drained Area | DA | $m^2$ | -0.004 | [-0.08 ; 0.05 ] |
| The Hydraulic Altitude | HAlt | m | 0.063 | [-0.04 ; 0.08] |
| HAND | HAND | - | -0.01 | [-0.09 ; 0.06 ] |